# A Complex Genomic Rearrangement Resulting in Loss of Function of *SCN1A* and *SCN2A* in a Patient with Severe Developmental and Epileptic Encephalopathy

**DOI:** 10.3390/ijms232112900

**Published:** 2022-10-26

**Authors:** Valeria Orlando, Silvia Di Tommaso, Viola Alesi, Sara Loddo, Silvia Genovese, Giorgia Catino, Licia Martucci, Maria Cristina Roberti, Marina Trivisano, Maria Lisa Dentici, Nicola Specchio, Bruno Dallapiccola, Alessandro Ferretti, Antonio Novelli

**Affiliations:** 1Laboratory of Medical Genetics, Translational Cytogenomics Research Unit, Bambino Gesù Children Hospital, IRCCS, 00146 Rome, Italy; 2Rare and Complex Epilepsy Unit, Department of Neuroscience, Bambino Gesù Children Hospital, IRCCS, 00146 Rome, Italy; 3Medical Genetics Unit, Bambino Gesù Children Hospital, IRCCS, 00146 Rome, Italy; 4Genetics and Rare Disease Research Division, Bambino Gesù Children Hospital, IRCCS, 00146 Rome, Italy

**Keywords:** array CGH, CGR, complex genomic rearrangements, chromothripsis, cryptic rearrangement, OGM, optical genome mapping, genome sequencing, DEE, developmental and epileptic encephalopathy, *SCN1A*, SCN2A

## Abstract

Complex genomic rearrangements (CGRs) are structural variants arising from two or more chromosomal breaks, which are challenging to characterize by conventional or molecular cytogenetic analysis (karyotype and FISH). The integrated approach of standard and genomic techniques, including optical genome mapping (OGM) and genome sequencing, is crucial for disclosing and characterizing cryptic chromosomal rearrangements at high resolutions. We report on a patient with a complex developmental and epileptic encephalopathy in which karyotype analysis showed a de novo balanced translocation involving the long arms of chromosomes 2 and 18. Microarray analysis detected a 194 Kb microdeletion at 2q24.3 involving the *SCN2A* gene, which was considered the likely translocation breakpoint on chromosome 2. However, OGM redefined the translocation breakpoints by disclosing a paracentric inversion at 2q24.3 disrupting *SCN1A*. This combined genomic high-resolution approach allowed a fine characterization of the CGR, which involves two different chromosomes with four breakpoints. The patient’s phenotype resulted from the concomitant loss of function of *SCN1A* and *SCN2A*.

## 1. Introduction

Complex genomic rearrangements (CGRs) include structural variants with two or more breakpoints that are not fully characterized by conventional G-banded karyotyping and FISH. CGRs are non-recurrent rearrangements mostly caused by microhomology-mediated recombination processes [1]. Recently, a new phenomenon, chromothripsis, was defined as the shattering and reshuffling of one or a few chromosome segments during a one-step catastrophic event, associated with the incomplete repair of double-strand breaks (DSBs) through non-homologous end-joining (NHEJ) [2]. Initially, this phenomenon was reported in cancer cells, in which tens to hundreds of genomic rearrangements have been acquired in a single catastrophic event [3], while other studies suggest a similar mechanism of NHEJ in both CGRs and cancer [4]. However, recent evidence argues that chromothripsis could occur in the germline and in the early stage of embryonic development, leading to stable and heritable CGRs [5,6].

A fine characterization of CGRs is crucial for the identification of gene disruption at breakpoints and for evaluating the clinical outcome. Karyotype and FISH analysis can provide important genomic information but are unable to provide a fine characterization of the rearrangements. A combined approach of genomic high-resolution methods overcomes these limits and provides clues to a deeper genotype–phenotype analysis.

Genome sequencing and optical genome mapping (OGM) analysis, a non-sequencing genome imaging tool, are innovative approaches providing high-resolution information about numerical and structural rearrangements.

We report on a female patient affected by developmental and epileptic encephalopathy (DEE) [7] in which OGM and genome sequencing detected a de novo complex genomic rearrangement, not fully resolved by standard techniques, explaining the clinical phenotype.

### Clinical Description

We describe an 8-year-old girl with an unremarkable perinatal history, born at term from non-consanguineous parents. Family history was negative. Her 6-year-old brother is healthy. The patient had a normal development up to the age of 16 months, when neurodevelopmental regression was noticed with worsening of vigilance, feeding difficulties, and asthenic body habitus, following a first episode of seizures, occurring in conjunction with fever. A cranial TC scan and electroencephalogram (EEG) were considered normal. Afterwards, she developed afebrile generalized tonic–clonic, focal to bilateral tonic–clonic and myoclonic seizures, associated with resistance to anti-seizure medications (ASMs), including valproate, clonazepam, levetiracetam, phenobarbital, clobazam, sulthiame, rufinamide, lacosamide, carbamazepine, ketogenic diet, vigabatrin, topiramate, cannabidiol, and phenytoine, which did not result in a significant benefit or worsening of epilepsy and development. She also experienced several episodes of status epilepticus.

The patient was firstly evaluated by us at the age of 5 years, when she manifested severe cognitive and intellectual disability with limited social interaction, poor eye contact, absent speech, behavior disturbances and moderate axial hypotonia and hyporeflexia. She experienced drug-resistant focal–tonic seizures with secondary bilateral clonic-tonic diffusion and myoclonic seizures, with multiple clusters per day. Video-EEG recording revealed a severe disruption of background activity with recurrent high voltage slow waves intermingled with epileptiform abnormalities, predominantly over the bilateral frontal and temporal regions. Ictal discharge showed a diffuse low-voltage fast activity, increasing in amplitude and decreasing in frequency, bilateral and symmetrical, associated with a massive and diffuse tonic contraction, with flushing of the face and trunk, perioral cyanosis, and sialorrhea, followed by tonic–clonic contraction of upper and lower limbs lasting 90 s (Appendix A). A clinical diagnosis of developmental and epileptic encephalopathy (DEE) was made.

## 2. Results

NGS analysis was performed on the patient’s blood as a first-line test, focusing on genes associated with epileptic encephalopathies, but no pathogenic variants were detected. Cytogenetic analysis revealed a female karyotype with a de novo balanced translocation involving the long arms of chromosomes 2 and 18 (Figure 1). Array-CGH analysis disclosed a de novo microdeletion at 2q24.3, spanning 194 Kb of genomic DNA (arr[GRCh37]2q24.3(166,079,043_166,273,000) × 1 dn), including the entire *SCN2A* gene (Figure 2), which was considered the likely translocation breakpoint on chromosome 2.

Genome sequencing and OGM confirmed the 2q24.3 deletion and the reciprocal 2q;18q translocation, but ruled out the association between the two rearrangements. In fact, the translocation breakpoint was found to map several Mb downstream in a gene “desert” region at 2q32.1. A paracentric inversion was detected within this region, with the distal breakpoint at 2q32.1, overlapping the translocation breakpoint and the proximal breakpoint at 2q24.3, 655 Kb downstream the deletion disclosed by CMA. Interestingly, the 2q24.3 proximal breakpoint disrupted the first intron of the *SCN1A* gene, probably causing its functional inactivation (Figure 3). The translocation breakpoint on chromosome 18, at 18q21.32, disrupted the *ALPK2* gene, whose haploinsufficiency is not associated with clinical outcome to date.

## 3. Discussion

The integrated use of high-resolution platform (CMA) and new technologies, including OMG and genome sequencing, proves to be helpful in the fine characterization of CGRs, which is mandatory in establishing accurate genotype–phenotype correlation. The present patient showed a complex phenotype of developmental and epileptic encephalopathy (DEE) with neurodevelopmental regression since 16 months of age, when the epilepsy started, with different seizures semiology and a strong resistance to ASMs.

Cytogenetic analysis revealed a balanced translocation involving the long arm of chromosomes 2 and 18, and CMA detected a small de novo 2q24.3 microdeletion, including the *SCN2A* gene. These results did not seem per se capable of explaining the rapid evolution and the complexity of clinical outcome, since recent studies report loss-of function variations of the *SCN2A* gene in association to intellectual disability (ID) without seizures [8].

Thus, OGM and genome sequencing were used for a deeper characterization of the rearrangement, with the aim of refining the correlation between the 2q24.3 microdeletion and the translocation breakpoints. Genome sequencing and OGM highlighted a CGR involving the 2q24.3 region, characterized by four breakpoints, two delimiting the known *SCN2A* gene deletion and two more distal, delimiting a paracentric inversion, including one overlapping the translocation breakpoint and one into the first intron of the *SCN1A* gene. The inversion causes the displacement of the 5′ UTR region and the first exon of the *SCN1A* gene from 2q24.3 to 2q32.1, leading to probable inactivation of the gene (Figure 4). To our knowledge, this is the first patient in which the concomitant inactivation of both the *SCN1A* and *SCN2A* genes has been documented.

*SCN1A* and *SCN2A* encode Nav1.1 and Nav1.2, respectively, two of nine α-subunits of the channel pore of voltage-gated sodium (Nav) channels, together with more β-subunits modulating function [9]. Nav channels have an essential role in correcting the neurological functions involved in the initiation and propagation of action potentials across the central nervous system [10]. *SCN1A* and *SCN2A* are the most commonly mutated epilepsy-associated genes, with different pathogenic variants leading to a wide range of phenotypes with variable disease severity [11]. More than 140 variants in *SCN1A* and more than 200 in *SCN2A* have been described [12].

Pathogenic variants in *SCN2A* are associated with a wide spectrum of neurodevelopmental disorders and four different phenotypes have been delineated, including benign familiar neonatal-infantile seizures (BFIS3) (OMIM # 607745), episodic ataxia type 9 (EA9) (OMIM #618924), developmental and epileptic encephalopathy-11 (DEE11) (OMIM #613721), autism spectrum disorder and intellectual disability (ASD/ID) [13,14].

Causative variants in *SCN1A* were identified in patients with an age-dependent epileptic encephalopathy, known as Dravet syndrome (DS) (OMIM # 607208), and in patients showing a spectrum of seizure disorders, ranging from early-onset isolated febrile seizures to generalized epilepsy with febrile seizures plus, type 2 (GEFSP2) (OMIM # 604403). Missense variants of *SCN2A* are the most common variants in epileptic encephalopathies and they typically have a gain-of-function effect (GOF), resulting in patients which usually benefit from sodium channel blockers (SCBs), whereas de novo loss of function variants inducing *SCN1A* haploinsufficiency are usually associated with DS [13].

The clinical characteristics of our patient do not exactly match with any specific phenotype classified in association with *SCN1A* or *SCN2A* gene variations. In the present patient, the epilepsy onset was after the first year of life, while in Dravet syndrome the onset of seizures typically is around 6 months of age, although a minority of cases manifest seizures in the second year of life. On the other hand, in the present case the first seizure was associated with a fever, and she experienced different seizure semiology (focal and generalized tonic–clonic and myoclonic seizures) as reported in Dravet syndrome, as well as psychomotor regression and behavior disorder [15].

Interstitial deletions of 2q24.3 are very rare. In 2015, Lim et al. [16] reviewed the literature of 2q24.3 deletions with variable extensions and, thus, differently involving the various sodium channel gene clusters (*SCN3A*, *SCN2A*, *SCN1A*, *SCN9A*, and *SCN7A*). *SCN1A* is considered the major contributor to the epileptic phenotype, although the role of other sodium channel genes that map within this cluster is less delineated. Patients with deletion of the entire sodium channel gene cluster exhibited a complex epilepsy phenotype characterized by migrating partial seizures of infancy with neonatal seizure onset, severe developmental delay and acquired microcephaly. Individuals with partial deletion of *SCN1A* and *SCN9A* and whole *SCN1A* deletion present with an epileptic phenotype of Dravet syndrome. Our patient presented a neurodevelopmental regression in conjunction with multiple drug-resistant afebrile generalized tonic–clonic and myoclonic seizures, followed by developmental and epileptic encephalopathy (DEE). The loss of function of *SCN1A* and the concurrent *SCN2A* gene deletion, due to cryptic CGRs likely arising from a chromotripsis event at germinal level or in the very early mitotic divisions, might explain her complex phenotype as result of a double hit. These results support the hypothesis that these events occur during an “all in one” chromosomal catastrophe rather than through the progressive acquirement of mutations or rearrangements.

In conclusion, the application of a combined genomic high-resolution approach allows the fine characterization of CGRs, and improves the molecular diagnosis leading to precise treatments and better clinical outcome.

## 4. Materials and Methods

### 4.1. Sample Collection and Consent

EDTA and Na-Heparin peripheral blood samples of the patient and her parents were collected and used for molecular and cytogenetic testing. DNA was isolated by means of a Qiagen blood kit (Qiagen, Hilden, Germany) according to the manufacturer’s instructions. The patient’s parents signed an informed consent for genetic analysis and publication purpose. The study was approved by the Ethics Committee of OPBG Hospital in compliance with the Helsinki Declaration.

### 4.2. Mutation Analysis

A custom gene panel including genes associated with epileptic encephalopathy was analyzed by next-generation sequencing (NGS) on genomic DNA. The patient’s library preparation and targeted resequencing were performed using the NimbleGen SeqCap Target Enrichment kit (Roche, Basilea, Switzerland) on a NextSeq550 (Illumina, San Diego, CA, USA) platform, according to the manufacture’s protocol. The BaseSpace pipeline (Illumina, San Diego, CA, USA) and the Geneyx Analysis software (Geneyx, Genomex Ltd., Herzliya, Israel) were used for the variant calling and annotating variants, respectively. Sequencing data were aligned to the hg19 human reference genome.

### 4.3. Cytogenetics Analysis

Karyotype analysis was performed on metaphases from lymphocyte cultures according to standard G-banding techniques.

### 4.4. Array

CMA (chromosomal microarray analysis) was performed on an array-CGH 4 × 180K platform (Agilent Technologies, Santa Clara, CA, USA) using standard procedures. Images were obtained using an Agilent DNA Microarray Scanner and analyses were performed by Agilent CytoGenomics (v 5.1.2.1). Confirmation and segregation tests on the patient’s and parents’ DNA were performed by Sybr Green qPCR [17] on the *SCN2A* gene with the *TERT* gene as internal control.

### 4.5. Optical Genome Mapping (OGM)

A fresh blood sample was collected in EDTA and stored at −80 °C just after sampling. Ultra-high molecular weight (UHMW) DNA was extracted according to manufacturer’s instructions (SP Frozen Human Blood DNA Isolation Protocol, Bionano Genomics, San Diego, CA, USA) and enzymatically labeled (Bionano Prep Direct Label and Stain Protocol). Labeled DNA was uploaded on nanochannel chips and scanned on a Saphyr instrument (Bionano Genomics). An effective genome coverage >100× was achieved. Images were analyzed by Access software v3.6, using Bionano De novo genome assembly pipeline. Genome maps obtained were aligned with Human Genome Reference Consortium GRCh38/hg38 assembly for structural variant detection.

### 4.6. Genome Sequencing

Library preparation was carried out according to the manufacturer’s protocol from DNA PCR-Free Library Prep (Illumina), and sequenced on a NovaSeq6000 (Illumina) platform. The obtained NGS assay presented a mean coverage of 35×, with Q30 bases around 87%. The TruSight Software Suite (Illumina) and the integrated DRAGEN platform and IGV software were used for alignment, variant calling and breakpoint data visualization. Sequencing data were aligned to the hg38 human reference genome.

## Figures and Tables

**Figure 1 ijms-23-12900-f001:**
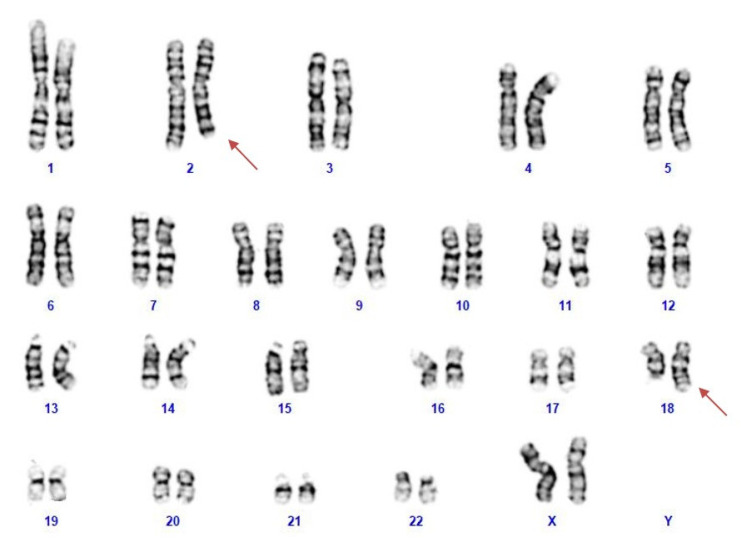
The patient’s karyotype showing a reciprocal translocation between the long arm of chromosomes 2 and 18.

**Figure 2 ijms-23-12900-f002:**
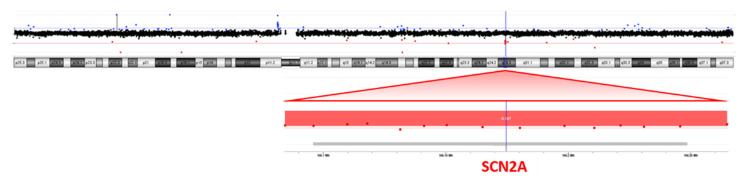
Array-CGH (4 × 180K platform) showing a microdeletion at 2q24.3, spanning 194 Kb of genomic DNA including the entire *SCN2A* gene.

**Figure 3 ijms-23-12900-f003:**
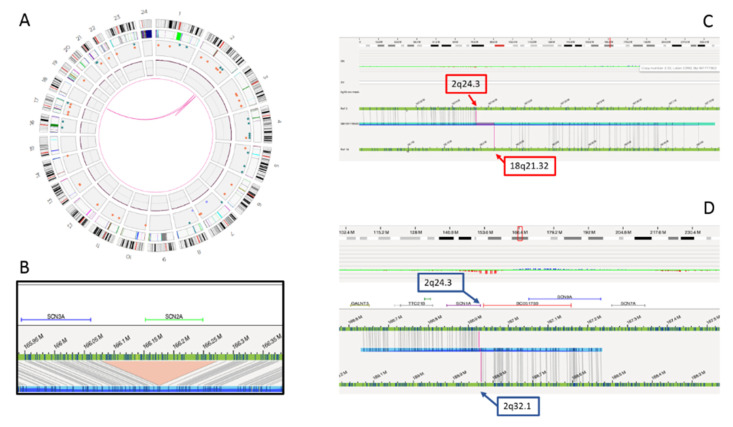
Complex genomic rearrangements (CGR) disclosed by optical genome mapping. Graphic visualizations of the rearrangement include the circle plot (**A**), the *SCN2A* gene deletion (**B**), the 2;18 translocation with breakpoints at 2q24.3 and 18q21.32 (**C**) and paracentric inversion involving 2q24.3 and 2q32.1 breakpoints (**D**).

**Figure 4 ijms-23-12900-f004:**
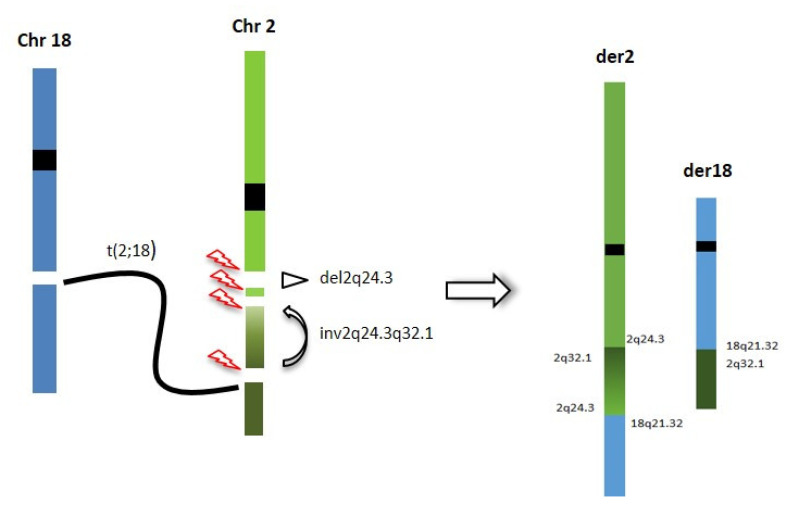
Chromotripsis event underlying the CGR identified in our patient: breakpoints are reported and chromosome segments reshuffling is represented.

## Data Availability

The data that support the findings of this study are available on request from the corresponding author. The data are not publicly available due to privacy or ethical restrictions.

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
