# Peer review of "A Complex Genomic Rearrangement Resulting in Loss of Function of SCN1A and SCN2A in a Patient with Severe Developmental and Epileptic Encephalopathy"

_ijms, 2022, doi:10.3390/ijms232112900_

Round 1

Reviewer 1 Report

The authors present an interesting case of a patient with developmental and epileptic encephalopathy wit combined SCN1A and SCN2A alterations. Overall the manuscript is well written.  I only have a few minor suggestions that I think could further enhance the manuscript:

1. Under the 'Clinical description' section, the authors describe video-EEG recordings. If possible, it would be helpful to include an illustration of these recordings. 

2.Under the 'Discussion' section, the authors briefly discuss 2q24.3 microdeletion. I believe it would be interesting if the authors could discuss the clinical phenotypes associated with this alteration as reported in the literature and how they differ from the patient in the case report. 

3. Briefly elaborating on the type of SCN1A and SCN2A mutations (e.g. gain-of-function, in-frame deletions) that can occur in association with epileptic encephalopathies. 

Author Response

Reviewer #1:

The authors present an interesting case of a patient with developmental and epileptic encephalopathy wit combined SCN1A and SCN2A alterations. Overall the manuscript is well written.  I only have a few minor suggestions that I think could further enhance the manuscript:

  1. Under the 'Clinical description' section, the authors describe video-EEG recordings. If possible, it would be helpful to include an illustration of these recordings. 

Accordingly, we provide an illustration of ictal video-polygraphic recording of our patient in “Supplementary Materials” section.

  1. Under the 'Discussion' section, the authors briefly discuss 2q24.3 microdeletion. I believe it would be interesting if the authors could discuss the clinical phenotypes associated with this alteration as reported in the literature and how they differ from the patient in the case report. 

As the reviewer suggested, we added in the “Discussion” section a paragraph with a more accurate description of the phenotypes associated with 2q24.3 deletions as reported in the literature.

  1. Briefly elaborating on the type of SCN1A and SCN2A mutations (e.g. gain-of-function, in-frame deletions) that can occur in association with epileptic encephalopathies.

As suggested, we added a brief description of the functional effects of mutations occurring in SNC1A e SCN2A.

Reviewer 2 Report

The authors report on an interesting case of complex genomic rearrangement assessed by genome sequencing and optical genome mapping. I have some minor comments to improve the manuscript:

- It is recommended to say genome sequencing and not whole genome sequencing

- Please use ISCN 2020 recommendations for the array result

- Figure 3 lacks the representation of the paracentric inversion as it is for the deletion (B) and the translocation (C). We see the paracentric inversion in (A) but I think it would be useful to see it also with the barcodes

- Authors state that the inversion disrupts the SCN1A gene leading to the gene inactivation. As functional experiments have not been done to confirm it, it seems reasonable to say “probable inactivation”

- Authors do not comment on the mechanism of both SCN1A and SCN1B inactivation. Can it be considered as digenism? Or is only a double hit?   

Author Response

Reviewer #2:

The authors report on an interesting case of complex genomic rearrangement assessed by genome sequencing and optical genome mapping. I have some minor comments to improve the manuscript:

  1. It is recommended to say genome sequencing and not whole genome sequencing

We made the changes asked by the reviewer, as suggested.

  1. Please use ISCN 2020 recommendations for the array result

We edited the array result according to ISCN 2020.

  1. Figure 3 lacks the representation of the paracentric inversion as it is for the deletion (B) and the translocation (C). We see the paracentric inversion in (A) but I think it would be useful to see it also with the barcodes

Accordingly, we modified Figure 3 adding an illustration of the paracentric inversion.

  1. Authors state that the inversion disrupts the SCN1A gene leading to the gene inactivation. As functional experiments have not been done to confirm it, it seems reasonable to say “probable inactivation”

We made the changes asked by the reviewer, as suggested.

  1. Authors do not comment on the mechanism of both SCN1A and SCN1B inactivation. Can it be considered as digenism? Or is only a double hit?   

As the reviewer suggested, we added a comment in “Discussion” section assuming a double hit mechanism.